# Preneoplastic Lesions in Surgical Specimens Do Not Worsen the Prognosis of Patients Who Underwent Surgery for Pancreatic Adenocarcinoma: Post-Hoc Analysis of the PRODIGE 24-CCTG PA 6 Trial

**DOI:** 10.3390/cancers14163945

**Published:** 2022-08-16

**Authors:** Théo Legrand, Julia Salleron, Thierry Conroy, Frédéric Marchal, Jacques Thomas, Laure Monard, James Jim Biagi, Aurélien Lambert

**Affiliations:** 1Department of Medical Oncology, Institut de Cancérologie de Lorraine, 54500 Vandœuvre-lès-Nancy, France; 2Biostatistic Unit, Institut de Cancérologie de Lorraine, 54500 Vandœuvre-lès-Nancy, France; 3Department of Surgical Oncology, Institut de Cancérologie de Lorraine, 54500 Vandœuvre-lès-Nancy, France; 4Department of Biopathology, Institut de Cancérologie de Lorraine, 54500 Vandœuvre-lès-Nancy, France; 5R&D UNICANCER, 75654 Paris, France; 6Department of Oncology, Queen’s University, Kingston, ON K7L 5P9, Canada

**Keywords:** pancreatic cancer, dysplasia, PanIN, IPMN, MCN, surgery

## Abstract

**Simple Summary:**

Pancreatic cancer patients who undergo curative surgery are still likely to recur. We therefore analyzed the data of the 493 patients from the PRODIGE 24-CCTG PA 6 trial, which validated the benefit of adjuvant mFOLFIRINOX regimen over gemcitabine after pancreatic adenocarcinoma resection. We investigated whether the presence of dysplasia (noninvasive intraductal papillary mucinous neoplasm, mucinous cystic neoplasm or pancreatic intraepithelial neoplasia) might decrease in disease-free survival. A preneoplastic lesion was identified in 226 patients (45.8%). In a multivariate analysis, the presence of dysplasia is not an independent predictor of diminished disease-free survival. This finding should be useful for future prospective trials and for surgeons’ decision making, as the pre-existence of a preneoplastic lesion should not preclude a plan for curative surgery.

**Abstract:**

Objective: The prognosis of pancreatic cancer after curative surgery is burdened by frequent recurrence. The aim of this study was to evaluate the impact of dysplasia in the surgical specimen on disease-free survival (DFS). Methods: A post-hoc analysis of the phase III PRODIGE 24-CCTG PA 6 trial was performed. From April 2012 to October 2016, 493 patients were included in the primary study. Assessment for dysplasia in the surgical specimens was secondarily performed. Dysplasia was defined based on presence and grade of three most common pre-malignant lesions (intraductal papillary mucinous neoplasm (IPMN), mucinous cystic neoplasm (MCN) and pancreatic intraepithelial neoplasia (PanIN). The primary endpoint was DFS validated through multivariate analysis. Results: Two hundred twenty-six patients (45.9%) had a preneoplastic lesion. PanIN lesions were found in 193 patients (39.2%), including 100 high-grade lesions (20.6%); 43 patients had IPMN lesions (8.7%), including high-grade lesions in 32 (6.5%). Three MCN were described (0.6%). In bivariate analysis, the presence of dysplasia was not associated with poorer DFS (HR = 0.82, 95% CI [0.66; 1.03]). In multivariate analysis, risk factors for poorer DFS were poorly differentiated/undifferentiated tumor, N1 status, R1 surgical margins and perineural invasion. Conclusions: The presence of dysplasia in the surgical specimen after pancreatic cancer surgery does not worsen DFS.

## 1. Introduction

Pancreatic cancer has become the third leading cause of death by cancer in men and women combined and is expected to be the second leading cause by 2040 [1]. Its mortality has increased slightly in men and remained relatively stable in women for the last 20 years [2]. The 5-year overall survival (OS) after curative pancreatic surgery remains low and leaves room for improvements. The PRODIGE 24-CCTG PA 6 clinical trial was designed to compare the efficacy and safety of mFOLFIRINOX (modified FOLFIRINOX) to gemcitabine as adjuvant treatment in patients with resected pancreatic ductal adenocarcinoma. This study demonstrated improved OS and disease-free survival (DFS) with mFOLFIRINOX at the cost of increased but manageable toxicity [3].

The three most common preneoplastic lesions (dysplasia) of the pancreas are: intraductal papillary mucinous neoplasm (IPMN), mucinous cystic neoplasms (MCN), and pancreatic intraepithelial neoplasia (PanIN) [4]. IPMNs are tumors that grow within the pancreatic ducts and produce mucin. IPMNs are fairly common and frequently present as incidental pancreatic cyst. The prevalence increase with age until it reaches in up to 10% of persons aged 70 years or older. Depending on their location in the ductal system, some of them progress to invasive cancer if they are left untreated [5,6]. PanIN are an important and well-known asymptomatic precursor of ductal pancreatic adenocarcinoma [7]. High-grade pancreatic intraepithelial neoplasia (previously named PanIN-3) are nearly exclusively adjacent to pancreatic adenocarcinoma or detected in patients with a family history of pancreatic cancer or a predisposing germline mutation [5,8]. MCNs are seen nearly exclusively in women and are usually (95% of cases) located in the body or tail of the pancreas [6].

Dysplasia within the resection specimen after a pancreatectomy for non-invasive IPMN is associated with 21% recurrence as new cyst, IPMN or pancreatic carcinoma [9]. Moreover, dysplasia at a resection margin was associated with a three-fold increased risk of recurrence (*p* = 0.02) after resection of non-invasive IPMN, even if no relationship between dysplasia and cancer was found [9]. These findings warrant further investigation as the recurrence rate differs depending on the status of the resected margin after surgery of invasive or non-invasive IPMN [10].

The presence of a preneoplastic lesion either on the tumor margin or within the parenchyma, following pancreatic cancer surgery, has never been explored and its impact on possible recurrence is unknown. The objective of this study was to evaluate whether the presence of dysplasia in resected specimens is a poor prognostic factor for DFS.

## 2. Materials and Methods

A post-hoc analysis based on data from the prospective phase III PRODIGE 24 CCTG PA 6 trial was performed. Inclusion criteria were: age from 18 to 79 years, histologically confirmed pancreatic ductal adenocarcinoma, complete macroscopic R0 [no cancer cells within 1 mm of all resection margins] or R1 [cancer cells present within 1 mm of one or more resection margins] resection within 3 to 12 weeks before randomization, no evidence of metastatic disease, malignant ascites, or pleural effusion. Other inclusion criteria were: full recovery from surgery, a World Health Organization (WHO) performance status score of 0 or 1, and adequate hematologic, liver, and renal function. Patients with non-ductal pancreatic tumors, incomplete (R2) resection, a serum CA 19-9 level of more than 180 kU/L within 21 days before randomization, previous chemotherapy or radiotherapy, or symptomatic heart failure or coronary heart disease were ineligible.

From April 2012 to October 2016, a total of 493 patients from 58 centers in France and 19 centers in Canada were randomly assigned to receive a planned 6-month course of either mFOLFIRINOX (247 patients) or gemcitabine (246 patients). Recommendations were made for surgeons and pathologists to orient the tumor specimen to accurately identify the resection limits. During surgery, pancreatic neck margin was checked by intraoperative frozen section. In case of positivity, an additional resection was advised until a negative margin was obtained. Patients had regular standardized surveillance, which included thoraco-abdominopelvic computed tomography scans or MRI, serum CA 19-9 levels, and clinical examinations repeated every 3 months for 2 years and then every 6 months for 3 years. Full treatment details have been published previously [3]. The database was locked on 13 April 2018, at which time 314 cancer-related events, second cancers, or deaths from any cause (91.8% of the expected DFS events) had occurred.

Our current study was based on 3-year data from the PRODIGE 24-CCTG PA 6 clinical trial, and all patients were analyzed.

Pathology reports of each patient were reviewed to determine the presence of a preneoplastic lesion (IPMN, MCN, PanIN or mixed), associated with the primary invasive tumor and its location, i.e., on the tumor resection margin, within the pancreatic parenchyma or both. The location of the lesions on the tumor resection margin was specified, i.e., on the main pancreatic duct or on the branch ducts. Grade of the dysplasia was classified as low/intermediate grade, high-grade or invasive according to the Baltimore consensus [11]. For parenchymal dysplastic lesions, the type of lesion and its grade were also classified according to the Baltimore consensus. When dysplasia was present at multiple locations within the pancreas with different grades, we classified considering the highest grade.

This study was declared to the French National Commission on Informatics and Liberties (CNIL) and received prior authorization from the study sponsor, Unicancer.

The primary endpoint was DFS based on the presence or absence of a preneoplastic lesion in the surgical sample. DFS was calculated from the date of randomization to the date of the first cancer-related event, second cancer or death from any cause. Patients without events at the time of analysis had their data censored on the date of last informative follow-up.

### Statistical Analysis

All statistical analyses were performed using SAS software, v9.4 (SAS Institute Inc., Cary, NC 25513, USA). Two-sided *p*-values < 0.05 were considered statistically significant. Qualitative parameters were described by frequency and percentage.

Predictive factors for DFS were assessed using bivariate Cox proportional hazards models adjusted by treatment arm [12]. The results were expressed as adjusted hazard ratio (HR) and 95% confidence interval. The validity of the proportional hazard assumption was checked using the Scaled Schoenfeld Residuals. Parameters with a *p*-value less than 0.2 were introduced in a full multivariate model. In line with the parsimony principle, this full model was then simplified with a backward selection.

Due to missing values on demographic and disease characteristics at inclusion, multiple imputations (MI) were performed according to FDA-approved methods, and we assumed that the described observations were missing at random [13]. Multiple pilot runs of various numbers of imputations were performed to assess the number of imputations and the stability of the parameter estimates for a given number of imputations. The number of 5 imputations was defined according to the fraction of missing information and the relative efficiency. The following method was applied: (a) multiple imputation using fully conditional specification was performed on 5 datasets using the proc mi in SAS; the MI model was performed on all demographic and disease characteristics. The DFS and time to events were also included. (b) For each imputed dataset, the effect of each predictive factors was assessed using bivariate Cox proportional hazards model adjusted on treatment arm. (c) The 5 estimates were then combined across imputed datasets with the proc mianalyze. As for complete case analysis, parameters with a *p*-value less than 0.2 were selected for the full multivariate analysis. For each imputed data set, we constructed 200 bootstrap data sets by randomly drawing with replacement. The total number of data sets was thus equal to 1000 data sets (5 (number of imputations) times 200 (number of bootstrap samples)). We then applied a backward selection on them and calculated the proportion of times that a variable appears in the final model namely the inclusion frequency. Parameters with an inclusion frequency greater than 0.7 were selected for the final multivariate model. The adjusted effect of each selected predictive factors was computed by combining the 5 estimates across imputed datasets with the proc mianalyze.

## 3. Results

Table 1 presents the demographic and disease characteristics of the 493 patients at baseline for the whole population. The majority were male (56.2%), younger than 65 years (59.1%) and of WHO performance status 0 (51.1%). Regarding comorbidities, 25.9% had diabetes mellitus and 46% were smokers.

Concerning histopathologic features, 98.8% of the patients had a ductal adenocarcinoma (98.8%), moderately (53.9%) or well (32.3%) differentiated, and location as head of the pancreas for 75.4%. Also, 88.6% of the tumors were classified as pT3 or pT4 and 76.5% as pN1, i.e., stage IIa or IIb in 91.7% of patients. The surgical margins were classified as R0 in 57.2% of patients. Lympho-vascular invasion was observed in 68.3% of cases and peri-neural invasion in 91.2%.

The rates of preneoplastic lesions are detailed in Table 2. Complete final pathology reports were available for 492 patients (99.8%). A preneoplastic lesion was found in 226 (45.9%) of resected tumors. In twenty cases (4.1%), a lesion was observed on the tumor margin only, in 161 (32.7%) in the pancreatic parenchyma only and in 45 (9.2%) the location was mixed, both on the surgical margin and the resected specimen. PanIN lesions were found in 193 patients (39.2%): 86 had low or intermediate grade lesions (17.7%), 100 had high-grade lesions (20.6%). A total of 7 additional patients with PanIN described were unspecified regarding the grade of differentiation.

IPMN lesions were present in 43 patients (8.7%), with low or intermediate grade for 10 (2%) and high-grade in 32 (6.5%). Three MCN were described (0.6%), one of which was intermediate grade and two were high-grade.

The median follow-up duration was 33.6 months, 95% CI [30.3; 36.0] and DFS at 3 years was 39.7%, 95% CI [32.8; 46.6] in the mFOLFIRINOX group (134 recurrences), as compared with 21.4% 95% CI [15.8; 27.5] in the gemcitabine group (180 recurrences, *p* < 0.001).

Pooling the 2 treatment arms, results of bivariate analyses adjusted on arm and after multiple imputations are presented Table 3. The presence of a preneoplastic lesion is not associated with a significant reduced DFS with a hazard ratio of 0.82, [95% CI 0.66; 1.03] (*p* = 0.088) (Figure 1). Regarding the location of these preneoplastic lesions, lesions on both tumor margin and parenchyma were associated with a better DFS (HR 0.58, [95% CI 0.37; 0.90], *p* = 0.015). IPMN lesions were also associated with a better DFS (HR 0.63, [95% CI 0.41; 0.97], *p* = 0.038), but no significant association was found for low/intermediate grade and high-grade separately. The DFS was not different according to the presence of PanIN lesions (HR 0.97, [95% CI 0.78; 1.22], *p* = 0.8).

An analysis was performed to determine whether the presence of precancerous lesions increased the risk of locoregional relapse. No association was found between the presence of preneoplastic lesions and locoregional relapse after adjustment for treatment arm (HR 0.82 [95% CI 0.59; 1.15], *p* = 0.252).

When analyzing other potential prognostic factors, higher tumor grade is associated with poorer DFS. DFS was also significantly associated with pN1 status (HR 1.76, [95% CI 1.32; 2.35]), stage IIa/IIb or III/IV (HR 2.09 [1.14; 3.81] and HR 6.62 [2.72; 16.14] respectively), surgical margins R1 (HR 1.47 [1.18; 1.84]) as well as venous resection (HR 1.41 [1.10; 1.80]), and in particular portal vein resection (HR 1.41 [1.06; 1.88]). The presence of lymphovascular invasion was also a poor predictive factor with a hazard ratio of 1.40 [1.09; 1.80] as well as perineural invasion (2.52 [1.57; 4.05]).

Including all predictive factors in a multivariate model (Table 4), with adjustment for treatment arm, the independent risk factors for DFS were poorly differentiated/undifferentiated tumor, pN1 nodal status, R1 surgical margins and perineural invasion. Location of preneoplastic lesions and the presence of IPMN lesions did not remain associated with DFS in the multivariate analysis. It should be noted that the presence or not of a dysplasia on the tumor margin was included as a risk factor in the analysis, but this had no impact on the DFS in multivariate analysis.

## 4. Discussion

We aimed to identify if the presence of preneoplastic lesions in resected pancreatic cancer specimens affect DFS of the patients included in the international PRODIGE 24-CCTG PA 6 study. In this population, we did not find a significant correlation between DFS and dysplasia at tumor margin or within the pancreatic specimen, neither on a preneoplastic lesion in the resected parenchyma, nor specifically according to the presence of an IPMN or a PanIN. It is worth noting the low number of MCN present, namely none on tumor margin and 0.6% on the rest of the pancreatic tissue.

Based on a large population and robust statistical analysis (multiple imputations combined with bootstrap resampling method), our study confirmed the main prognostic factors already known [3,14,15,16,17,18,19,20,21]. The most important pejorative factors seem to be on the one hand the histological characteristics of the primary tumor, in particular poorly differentiated or undifferentiated grade, pN1 status, presence of capsular rupture, perineural invasion and stage II-III-IV and, on the other hand, surgical factors, including invasive carcinoma on a surgical margin (R1 status) and the need for venous resection.

The rate of IPMN found in our study (8.7% of patients) is consistent with the data in the literature [22,23,24]. Counterintuitively, the presence of IPMNs associated with carcinoma rather shows a protective effect on DFS. One explanation for this significance in bivariate analysis is the fact that there is more pN0 status and overall, less risk factors in patients with IPMN than without (data not shown). This would come from the fact that a proportion of these patients may have been monitored in a context of known cyst lesion and had surgery for malignant IPMN. Other series also suggest that pancreatic ductal adenocarcinomas derived from IPMN and those concomitant with IPMN are significantly smaller, less invasive, and less extensive than ordinary pancreatic ductal adenocarcinoma [25,26]. Two series undergoing resection for IPMN-associated carcinoma indicated that prognosis of IPMN-associated carcinoma depends on histologic tumor subtype and grading [27,28]. A series of 35 patients having pancreatic resection for cancer aimed to assess the impact of the preneoplastic lesion pattern on survival. The authors reported that PanIN-related carcinomas displayed more aggressive features than IPMN-related carcinomas. Survival was significantly lower in cancers arising from PanIN preneoplastic lesions [27]. In a large series of 424 patients of IPMN-associated carcinomas, IPMN-ductal adenocarcinomas were diagnosed at more advanced tumor stages, more frequently involved lymph nodes, poorer differentiation and positive-margin resection and were associated with shorter median overall survival than IPMN-colloid carcinoma (26.7 months vs. 91.3 months) [28].

In the extensive review of the expert working group of the International Association of Pancreatology, the mean frequency of invasive IPMNs is 18.5% (6.1–37.7%) in branch duct IPMNs versus 43.1% (11–81%) in main duct IPMNs [29]. So, surgical treatment is recommended in all cases of IPMN with involvement of the main pancreatic duct and if there are high risk factors for malignancy [29,30,31] or worrisome features with involvement of the branch ducts [32]. Similarly, despite a lower risk of invasive carcinoma (about 13%), surgery is indicated for MCN ≥ 40 mm and patients who are symptomatic or have risk factors, such as mural nodule [7,31]

To our knowledge, few studies examined the prognostic impact of dysplasia on resected invasive pancreatic cancer. Frankel et al. addressed this question after pancreatic surgery in a series of 192 patients with non-invasive IPMN [9]. The presence of PanIN on the surgical margin was associated with a nearly doubled rate of recurrence at the remnant pancreas (*p* = 0.02). When any dysplasia (IPMN or PanIN) was present, the recurrence rate was 31% compared with 12% when the margin was not involved (*p* = 0.002). This was associated with a significantly decreased DFS (*p* = 0.001) without significant impact on OS. The meta-analysis by Leng et al. examined the impact of marginal status and lymph node involvement after surgery for invasive and non-invasive IPMN [10]. Out of 11 studies including 339 patients with invasive IPMNs, the incidence of recurrence with invasive IPMNs was 33.9% when margins were negative and 53.7% with positive margins (OR = 0.47; 95% CI: 0.25–0.88, *p* = 0.02). This significant unfavorable impact on recurrence when the margins were positive was also detected for non-invasive IPMNs: in 701 patients from 12 studies, the recurrence rate in patients with noninvasive IPMNs was 3.7% when margins were negative and 9.6% when margins were positive, a significant difference (OR = 0.37, 95% CI: 0.17–0.78, *p* = 0.01). Therefore, they recommended that the R0 surgery designation include margin dysplasia. In our study, we did not find any impact on locoregional relapse, but it should be noted that our patients had resection for pancreatic adenocarcinoma and received adjuvant chemotherapy. In the PRODIGE 24 trial, the median DFS in all patients was 15.8 months [14.2–18.9] [3]. Locoregional recurrence was the first event in 22.6% of the patients, locoregional plus distant recurrence occurred in 21% and distant recurrence occurred in 48.4%. This indicates that pancreatic cancer can be regarded as a systemic disease despite resection and adjuvant chemotherapy. This may explain our results: the poor prognosis of the pancreatic cancer largely outweighs the risk of local relapse linked to a preneoplastic lesion. Complementary work on the invasion of the margins by an invasive tumor is in progress to clarify the impact of each positive margin on the risk of local recurrence, in particular after Whipple resection.

One limitation of our trial is the short follow-up of 33.6 months. However, in the PRODIGE 24 updated 5-year analysis [33], local recurrence as first event occurred early at a median of 12.4 months (95% CI, 9.5–15.2 months), with no difference between local and metastatic relapse (10.2 months; 95% CI, 9.3–13.7), and we do not think that a longer follow-up should change these data.

The strength of our results lies in the large cohort of patients from a prospective, multicenter and international trial [3]. One limitation is that our conclusions are not based on OS but only on DFS. Even if it is not a validated surrogate endpoint for OS, it provides an earlier and robust outcome, especially for pancreatic cancers where death is both early and frequent.

## 5. Conclusions

In this post-hoc analysis of the PRODIGE 24 CCTG PA 6 trial, the presence of dysplasia (TIPMP, MCN or PanIN) in the surgical specimen, regardless of its grade of differentiation and location, has no deleterious effect on DFS in patients operated for pancreatic cancer and receiving adjuvant chemotherapy.

## Figures and Tables

**Figure 1 cancers-14-03945-f001:**
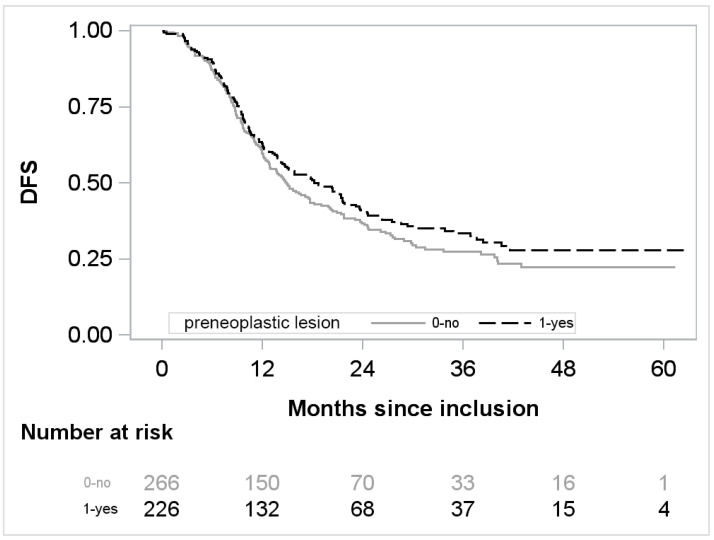
DFS over time regarding the presence or absence of preneoplastic lesion.

**Table 1 cancers-14-03945-t001:** Demographic and clinical characteristics of the patients at baseline.

Characteristics	No. (%)
Study arm	
Gemcitabine	246/493 (49.9)
mFOLFIRINOX	247/493 (50.1)
Age (years) ≥ 65	201/493 (40.8)
Male sex	277/493 (56.2)
WHO performance-status score	
0	249/487 (51.1)
1	238/487 (48.9)
Diabetes mellitus	126/487 (25.9)
Location of the tumor	
Head of the pancreas	372/492 (75.6)
Other	120/492 (24.4)
Tumor histologic findings	
Ductal adenocarcinoma	486/492 (98.8)
Nonductal adenocarcinoma	6/492 (1.2)
Tumor grade	
Well differentiated	149/462 (32.2)
Moderately differentiated	249/462 (54.0)
Poorly differentiated or undifferentiated	64/462 (13.8)
Primary tumor status	
pT1 or pT2	56/493 (11.4)
pT3 or pT4	437/493 (88.6)
Lymph node status	
pN0	116/493 (23.5)
pN1	377/493 (76.5)
Tumor stage	
IA or IB	26/493 (5.3)
IIA or IIB	452/493 (91.7)
III or IV	15/493 (3.0)
Surgical margins	
R0	282/493 (57.2)
R1	211/493 (42.8)
Lymphovascular invasion	289/423 (68.3)
Perineural invasion	412/452 (91.2)
Capsular rupture	75/432 (17.4)
Vascular resection	
Venous resection	122/490 (24.9)
Portal vein resection	74/493 (15.0)
Superior mesenteric vein resection	44/493 (8.9)
Arterial resection	15/492 (3.1)
Postoperative CA 19-9 level	
≤90 U/mL	457/493 (92.7)
>90 U/mL	36/493 (7.3)
Results presented as no./total no. (%)	

**Table 2 cancers-14-03945-t002:** Description of dysplastic lesions.

	All N = 493	Gemcitabine Arm N = 246	mFOLFIRINOX Arm N = 247
Preneoplastic lesion (n^0^, %)	226/492 ^α^ (45.9)	118/246 (48)	108/246 ^α^ (43.9)
Location of preneoplastic lesions			
Tumor margin	20/492 (4.1)	12/246 (4,9)	8/246 (3.3)
Parenchyma	161/492 (32.7)	82/246 (33,3)	79/246 (32.1)
Tumor margin and parenchyma	45/492 (9.2)	24/246 (9.8)	21/246 (8.5)
IPMN lesions	43/492 (8.7)	27/246 (11.0)	16/246 (6.5)
Grading IPMN			
Low- or intermediate-grade	10/491 (2)	7/245 (2.9)	3/246 (1.2)
High-grade	32/491 (6.5)	19/245 (7.8)	13/246 (5.3)
PanIN lesions	193/492 (39.2)	97/246 (39.4)	96/246 (39.0)
Grading PanIN			
Low- or intermediate-grade	86/485 (17.7)	41/242 (16.9)	45/243 (18.5)
High-grade	100/485 (20.6)	52/242 (21.5)	48/243 (19.8)
MCN lesions	3/492(0.6)	1/246 (0.4)	2/246 (0.8)
Grading MCN			
Low- or intermediate-grade	1/492 (0.2)	1/246 (0.4)	0
High-grade	2/492 (0.4)	0	2/246 (0.8)

Results presented as no./total no. (%). ^α^ Missing data for one patient regarding the preneoplastic lesions. Abbreviations: IPMN: intraductal papillary mucinous neoplasm; MCN: mucinous cystic neoplasms; PanIN: pancreatic intraepithelial neoplasia.

**Table 3 cancers-14-03945-t003:** Bivariate analyses adjusted on treatment arm (mFOLFIRINOX regimen or gemcitabine) after multiple imputations.

Characteristics		HR and 95% CI	*p*-Value
Age	≥65 vs. <65	1 [0.8; 1.25]	1
Gender	Male vs. Female	1 [0.80; 1.25]	1
WHO performance status score	1 vs. 0	1.09 [0.88; 1.37]	0.4
Diabetes mellitus	Yes vs. No	1.11 [0.86; 1.42]	0.4
Tumor location	Head vs. Other	1.13 [0.87; 1.47]	0.3
Tumor histologic findings	Ductal adenocarcinoma	1	
	Nonductal carcinoma	0.94 [0.35; 2.54]	0.9
Tumor grade	Well differentiated	1	
	Moderately differentiated	1.3 [1.01; 1.68]	0.043
	Poorly differentiated or undifferentiated	1.79 [1.26; 2.55]	0.001
Primary tumor pT stage	pT3 or pT4 vs. pT1 or pT2	1.28 [0.89; 1.84]	0.18
Nodal status	pN1 vs. pN0	1.76 [1.32; 2.35]	<0.001
Tumor stage	IA or IB	1	
	IIA or IIB	2.09 [1.14; 3.81]	0.017
	III or IV	6.62 [2.72; 16.14]	<0.001
Status of surgical margins	R1 vs. R0	1.47 [1.18; 1.84]	<0.001
Lymphovascular invasion	Yes vs. No	1.40 [1.09; 1.80]	0.009
Perineural invasion	Yes vs. No	2.52 [1.57; 4.05]	<0.001
Capsular rupture	Yes vs. No	1.47 [1.10; 1.95]	0.008
Venous resection	Yes vs. No	1.41 [1.10; 1.80]	0.007
Portal-vein resection	Yes vs. No	1.41 [1.06; 1.88]	0.020
Superior-mesenteric vein resection	Yes vs. No	1.44 [0.99; 2.10]	0.053
Arterial resection	Yes vs. No	0.78 [0.40; 1.52]	0.5
Postoperative CA 19-9 level	>90 U/mL vs. ≤90 U/mL	1.39 [0.93; 2.1]	0.11
Preneoplastic lesion	Yes vs. No	0.82 [0.66; 1.03]	0.088
Location of preneoplastic lesions	Tumor margin	1.02 [0.58; 1.80]	0.9
	Parenchyma	0.88 [0.69; 1.12]	0.3
	Tumor margin and parenchyma	0.58 [0.37; 0.90]	0.015
IPMN lesions	Yes vs. No	0.63 [0.41; 0.97]	0.038
Grading IPMN	Low or intermediate grade	0.64 [0.26; 1.58]	0.4
	High-grade—Invasive	0.63 [0.38; 1.03]	0.064
PanIn lesions	Yes vs. No	0.97 [0.78; 1.22]	0.8
Grading PanIn	Low or intermediate grade	1.00 [0.74; 1.36]	1
	High-grade	1.04 [0.72; 1.26]	0.7

Abbreviations: IPMN: intraductal papillary mucinous neoplasm; PanIN: pancreatic intraepithelial neoplasia; HR hazard ratio; 95% CI: 95% confidence interval.

**Table 4 cancers-14-03945-t004:** Multivariate analyses after multiple imputations.

	Full Multivariate Model		Final Multivariate Model *	
	HR and 95% CI	*p*-Value	HR and 95% CI	*p*-Value
Arm				
Gemcitabine	1		1	
mFOLFIRINOX	0.54 [0.43; 0.69]	<0.001	0.56 [0.45; 0.71]	<0.001
Tumor grade				
Well differentiated	1		1	
Moderately differentiated	1.2 [0.93; 1.55]	0.164	1.22 [0.94; 1.57]	0.131
Poorly differentiated or undifferentiated	1.86 [1.28; 2.71]	0.001	1.8 [1.26; 2.57]	0.001
Nodal status				
pN0	1		1	
pN1	1.38 [1.01; 1.89]	0.042	1.46 [1.08; 1.97]	0.014
Status of surgical margins				
R0	1		1	
R1	1.31 [1.04; 1.65]	0.021	1.33 [1.06; 1.67]	0.013
Lymphovascular invasion				
No	1			
Yes	1.1 [0.84; 1.45]	0.468	-	
Perineural invasion				
No	1			
Yes	1.9 [1.16; 3.12]	0.011	1.98 [1.21; 3.23]	0.006
Location of preneoplastic lesions				
Tumor margin	0.91 [0.5; 1.66]	0.757	-	
Parenchyma	0.86 [0.66; 1.11]	0.247	-	
Tumor margin and parenchyma	0.61 [0.38; 0.98]	0.041	-	
IPMN lesions				
No	1			
Yes	0.95 [0.59; 1.53]	0.838	-	

* After backward selection with bootstrap resampling method on each imputed data set. Abbreviations: IPMN intraductal papillary mucinous neoplasm; P; HR hazard ratio; 95% CI: 95% confidence interval.

## Data Availability

The data can be shared up on request.

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
