# Peer review of "Preneoplastic Lesions in Surgical Specimens Do Not Worsen the Prognosis of Patients Who Underwent Surgery for Pancreatic Adenocarcinoma: Post-Hoc Analysis of the PRODIGE 24-CCTG PA 6 Trial"

_cancers, 2022, doi:10.3390/cancers14163945_

Round 1

Reviewer 1 Report

We often come across the presence of pre-neoplastic lesions on the operative specimen of patients operated for pancreatic adenocarcinoma. The prognosis of these patients has not been adequately investigated and pancreatic recurrence is frequent.

In the same way, the finding of a positive surgical margin for pre-neoplastic lesion (High grade dysplasia) places the surgeon in front of a difficult choice already in the operating room.

The paper presented by Théo Legrand et al is full of interesting ideas. It is well written and supported with adequate statistical analysis.

Author Response

Review

Answer

Modification in text

#1

We often come across the presence of pre-neoplastic lesions on the operative specimen of patients operated for pancreatic adenocarcinoma. The prognosis of these patients has not been adequately investigated and pancreatic recurrence is frequent.

In the same way, the finding of a positive surgical margin for pre-neoplastic lesion (High grade dysplasia) places the surgeon in front of a difficult choice already in the operating room.

The paper presented by Théo Legrand et al is full of interesting ideas. It is well written and supported with adequate statistical analysis.

We thank you for your comments and suggestions which have strengthened the manuscript Please find below our answers to each comment.

cf. the entire manuscript

English language and style are fine/minor spell check required

A complete revision of the article for English has been done.

cf. the entire manuscript

Reviewer 2 Report

Authors stated that the presence of dysplasia on the surgical sample after pancreatic cancer surgery  does not worsen DFS.

Reviewer had some questions.

1. In this study, authors included all patients with GEM or mFOLFIINOX as adjuvant chemotherapy. Authors should show the data of each group with or without preneoplastic lesion.

2. Authors should compare the slice positive patients, because which means remnant pancreas still has preneoplastic lesions. 

3. In case of presence of preneoplastic lesion in MPD, did authors perform additional resection of pancreas?

4. In this study, authors utilized rapid pathological diagnosis for pancreas stump? Please describe about this.

5. Generally, if new cystic lesion occurs, the patients are supposed to receive careful management. Thus, reviewer thinks that DFS with positive will be differ little from  with negative preneoplastic lesions in his own opinion.  

Author Response

Review

Answer

Modification in text

#2

Authors stated that the presence of dysplasia on the surgical sample after pancreatic cancer surgery does not worsen DFS.

Reviewer had some questions.

1. In this study, authors included all patients with GEM or mFOLFIINOX as adjuvant chemotherapy. Authors should show the data of each group with or without preneoplastic lesion.

Thank you for this interesting suggestion. As it should have been, we have modified Table 2 to detail the data for each treatment arm, and it is indeed much more readable.

We replaced the entire Table 2. In the result section.

 See Table 2. Description of dysplastic lesions

2. Authors should compare the slice positive patients, because which means remnant pancreas still has preneoplastic lesions. 

This was not clearly explained but we performed a multivariate analysis taking this criterion into account.

In Table 4 we present the results of this analysis.

We modify the text of the article to explain it in a more evident way.

We added in the Results section:

It should be noted that the presence or not of a dysplasia on the tumor margin was included as a risk factor in the analysis, but this had no impact on the DFS in multivariate analysis.

3. In case of presence of preneoplastic lesion in MPD, did authors perform additional resection of pancreas?

An additional resection was planned in case of positive neck transection frozen section, but only in case of invasive tumor. In case of presence of a preneoplastic lesion on the main pancreatic duct, the study did not plan additional resection, so it was not performed. To date, this indication is under discussion in multidisciplinary staff, the benefit-risk ratio of a new resection invites us to be cautious because the benefit is unknown and the risk not null.

None.

4. In this study, authors utilized rapid pathological diagnosis for pancreas stump? Please describe about this.

Thanks for this important question. Yes, we did and the protocol asked for a perioperative frozen section on the pancreatic transection margin.

We added in the Material and methods section: Recommendations were made for surgeons and pathologists to orient the tumor specimen to accurately identify the resection limits. During surgery, pancreatic neck margin was checked by intraoperative frozen section. In case of positivity, an additional resection was advised until a negative margin was obtained.

5. Generally, if new cystic lesion occurs, the patients are supposed to receive careful management. Thus, reviewer thinks that DFS with positive will be differ little from with negative preneoplastic lesions in his own opinion.

Thank you for your pertinent comment, this is exactly what we tried to determine with this study. In conclusion, although management may be different and therefore have a different impact depending on the presence of preneoplastic lesions or not, this did not emerge significantly in our analysis, suggesting that this impact is minimal.

Our Discussion section has been modified to reflect this finding more explicitly.

Moderate English changes required

A complete revision of the article for English has been done.

cf. the entire manuscript.

Reviewer 3 Report

In this manuscript, the authors investigate the eventual association between the presence of pre-neoplastic lesions (PanIN, IPMN, and MCN) in the specimen (pancreatic parenchyma or surgical margin) and DFS among 493patients who underwent a pancreatic resection for pancreatic cancer (in the majority of cases for PDAC) and were who were enrolled  in the prodige-24 trial.  

Among 493 included patients, 226 patients had a pre-neoplastic lesion. The diagnosis of a pre-neoplastic lesion in the specimen was not associated with a reduced DFS.

The manuscript is well written, however some comments are due: 

- few misspellings and typos can be found in the manuscript: please carefully review it and correct. 

- the patients included in the study were operated between 2012 and 2016, the database was locked in 2018, leading to a median follow-up of 33 months. A follow-up duration extension may increase validity and robustenss of the study results.  

the discussion is quite short and limited: I suggest the authors to comment more in deep the results of their study and to better try to explain their findings. 

Author Response

Review

Answer

Modification in text

#3

In this manuscript, the authors investigate the eventual association between the presence of pre-neoplastic lesions (PanIN, IPMN, and MCN) in the specimen (pancreatic parenchyma or surgical margin) and DFS among 493patients who underwent a pancreatic resection for pancreatic cancer (in the majority of cases for PDAC) and were who were enrolled in the prodige-24 trial.  

Among 493 included patients, 226 patients had a pre-neoplastic lesion. The diagnosis of a pre-neoplastic lesion in the specimen was not associated with a reduced DFS.

The manuscript is well written, however some comments are due: 

- few misspellings and typos can be found in the manuscript: please carefully review it and correct. 

A complete revision of the article for English has been done.

cf. the entire manuscript.

- the patients included in the study were operated between 2012 and 2016, the database was locked in 2018, leading to a median follow-up of 33 months. A follow-up duration extension may increase validity and robustness of the study results.  

We agree with this remark, nevertheless the 5-year survival data are not yet published and will be the subject of a dedicated paper. Local recurrence as first event occurred early at a median of 12.4 months (95% CI, 9.5-15.2 months), with no difference with local and metastatic relapse (10.2 months; 95% CI, 9.3-13.7) and we do not think that a longer follow up will change these data.

We added in the Discussion section:

One limitation of our trial is the short follow-up of 33.6 months. However, in the PRODIGE 24 updated 5-year analysis [12], local recurrence as first event occurred early at a median of 12.4 months (95% CI, 9.5-15.2 months), with no difference with local and metastatic relapse (10.2 months; 95% CI, 9.3-13.7) and we do not think that a longer follow-up should change these data.

the discussion is quite short and limited: I suggest the authors to comment more in deep the results of their study and to better try to explain their findings. 

Discussion has been amplified, including explanation of our findings.

…/…

To explain our results, we added in the discussion:

In the PRODIGE 24 trial, the median DFS in all patients was 15.8 months [14.2-18.9] [2]. Locoregional recurrence was the first event in 22.6% of the patients, locoregional plus distant recurrence occurred in 21% and distant recurrence 48.4%. This indicates that pancreatic cancer can beregarded as a systemic disease despite resection and adjuvant chemotherapy. This may explain our results: the poor prognosis of the pancreatic cancer largely outweighs the risk of local relapse linked to a preneoplastic lesion. Complementary work on the invasion of the margins by an invasive tumor is in progress to clarify the impact of each positive margin on the risk of local recurrence, in particular after Whipple resection.

Reviewer 4 Report

This is a well written article, which adresses a controversial and important topic about pancreatic cancer. Data are retrieved from a prospective trial which changed clinical practice in PDAC treatment and data are clear, well represented and sustained by robust statistical analysis.

I've just some minor issues that could improve the manuscript, if applicable.

- Is there any correlation between presence of preneoplastic lesions and time of recurrence (early < 12 months vs late >24months)?

- Is there any correlations or difference between preneoplastic lesions presence in surgical specimens/margins and recurrence in pancreatic remnant vs distant metastases?

- Number of recurrences should be reported on Table 1 (also pooled for treatment arm);

Author Response

Review

Answer

Modification in text

#4

This is a well written article, which adresses a controversial and important topic about pancreatic cancer. Data are retrieved from a prospective trial which changed clinical practice in PDAC treatment and data are clear, well represented and sustained by robust statistical analysis.

I've just some minor issues that could improve the manuscript, if applicable.

- Is there any correlation between presence of preneoplastic lesions and time of recurrence (early < 12 months vs late >24months)?

Thank you for this comment, the analyses performed allow to answer this very relevant question due to the absence of impact of preneoplastic lesions. To illustrate this, we propose to add a Figure that describes the DFS according to the presence of a preneoplastic lesion which shows the absence of significant difference over time (it is understood that whether the relapse occurs before 12 months or after 24 months does not change the prognosis).

We added in the Results section:

The presence of a preneoplastic lesion is not associated with a significant reduced DFS with a hazard ratio of 0.82, [95% CI 0.66; 1.03] (p=0.088) (Figure 1).

- Is there any correlations or difference between preneoplastic lesions presence in surgical specimens/margins and recurrence in pancreatic remnant vs distant metastases?

Thanks for this important comment. Yes, we did the analysis and we found no impact of the presence of preneoplastic lesions on locoregional relapse after adjustment for treatment arm

We added in the Results section:

An analysis was performed to determine whether the presence of precancerous lesions increased the risk of locoregional relapse. No association between the presence of preneoplastic lesions and locoregional relapse after adjustment for treatment arm (HR 0.82 [95% CI 0.59; 1.15], p=0.252).

- Number of recurrences should be reported on Table 1 (also pooled for treatment arm);

Table 1 describes initial characteristics and not during follow-up, so to remain consistent with the presentation we propose to add the overall number of recurrences in the results section. In addition, Figure 1 now provides information on the number of events over time in a pooled analysis of the 2 treatment arms.

We added in the Results section:

 The median follow-up duration was 33.6 months, 95% CI [30.3;36.0] and DFS at 3 years was 39.7% 95% CI [32.8;46.6] in the modified-FOLFIRINOX group (134 recurrences), as compared with 21.4%95% CI [15.8;27.5] in the gemcitabine group (180 recurrences, p<.001).

English language and style are fine/minor spell check required 

A complete revision of the article for English has been done.

cf. the entire manuscript.

Round 2

Reviewer 2 Report

Their manuscript was improved.